# Optimizing Procedures for Antioxidant Phenolics Extraction from Skin and Kernel of Peanuts with Contrasting Levels of Drought Tolerance

**DOI:** 10.3390/foods11030449

**Published:** 2022-02-03

**Authors:** Adna P. Massarioli, Alan G. de O. Sartori, Fernanda F. Juliano, Roseane C. dos Santos, Jean Pierre C. Ramos, Liziane Maria de Lima, Severino Matias de Alencar

**Affiliations:** 1Department of Agri-Food Industry, Food and Nutrition, Luiz de Queiroz College of Agriculture, University of São Paulo, Piracicaba CEP 13418-900, Brazil; adnaprado@usp.br (A.P.M.); alangosartori@usp.br (A.G.d.O.S.); fernanda_ffj@hotmail.com (F.F.J.); 2Brazilian Agricultural Research Corporation (Embrapa Algodão), Campina Grande CEP 58428-095, Brazil; roseane.santos@embrapa.br (R.C.d.S.); jean.jp31@gmail.com (J.P.C.R.); liziane.lima@embrapa.br (L.M.d.L.)

**Keywords:** water stress, groundnut, reactive oxygen species, flavonoids, response surface methodology

## Abstract

Peanut is an affordable legume known for its nutritional value and phenolic content. The kernel and skin of 14 peanut genotypes contrasting in drought tolerance had their phenolic profiles determined and reactive oxygen species (ROS) scavenging activity evaluated. Firstly, temperature and % EtOH to extract antioxidant phenolic compounds were optimized using response surface methodology (RSM). The optimized extraction conditions, 60 °C and 35% EtOH for kernels and 40 °C and 60% EtOH for skins, were further adopted, and phenolic compounds were identified and quantified using high-performance liquid chromatography coupled with electrospray ionization-quadrupole-time of flight-mass spectrometry (HPLC-ESI-QTOF-MS) and high-performance liquid chromatography with photodiode array detector (HPLC-PDA). As a result, phenolic acids and glycosidic/non-glycosidic flavonoids were found. Principal component analysis was conducted, and the pairwise score plot of the skin extracts based on individual phenolic compounds showed a trend of genotype clustering based not only on drought tolerance but also on botanical type of germplasm. Therefore, our results demonstrate the status quo for antioxidant phenolic compounds of peanut genotypes contrasting in drought tolerance grown under natural field conditions.

## 1. Introduction

Peanut (*Arachis hypogaea* L.) is an affordable legume known for its nutritional value, mainly due to the high contents of protein, unsaturated fatty acids, and minerals [1]. Additionally, peanut has been widely recognized for its high content of phenolic compounds with antioxidant activity, which are concentrated in the seed skins and are related to health effects in the human body [2,3,4].

In Brazil, Runner cultivars (*hypogaea* germplasm), such as the Virginia botanic type, are widely grown because they are quite productive, although they are more sensitive to environments with water scarcity [5,6]. Peanut plants are known for presenting physiological and anatomical adjustments in response to water stress, although long periods of drought hamper productivity [6,7]. For environments with semiarid climate, where water supply is frequently irregular throughout the year, peanut can be a viable crop option due to the broad genetic variability of cultivars for drought tolerance, especially the *fastigiata* germplasm, represented by the Valencia and Spanish botanic types [5,6].

Oxidative stress is among one of the major biochemical changes caused by water stress. In response, different regulatory pathways, some involving the biosynthesis of secondary metabolites, work together to scavenge the excessive formation of reactive oxygen species (ROS), which cause damages in primary metabolites and in DNA [8]. In peanut, studies have demonstrated that the drought tolerant genotypes have a differentiated ability to scavenge ROS during water scarcity, and one of them is to activate the antioxidant system in order to prevent cell damage [9,10]. In this context, phenolic compounds are ubiquitous plant secondary metabolites with antioxidant activity, and modifications on the total content of phenolic compounds and on phenolic profile have been reported for whole (kernel + skin) seeds in plants tolerant to water stress, even when they were grown under conditions with no water stress [2,11].

The phenolic profile of peanut depends on the botanic type, germplasm, growth stage, water supply, and processing, as well as on how those compounds are extracted [4,11,12,13,14]. However, little is known regarding the effect of drought tolerance on phenolic profile and antioxidant activity in different edible parts of peanut genotypes. It is noteworthy that peanut skins, containing mainly condensed tannins such as procyanidins, are industrial by-products that have the potential to be sources of secondary metabolites applied in several fields, such as medicine, nutrition, and cosmetics [4,8]. The skin represents 2–7% of total peanut seed mass, while the kernel represents >90%, although the skin contains >90% of the phenolic compounds found in the seed [15]. Concerning market size, peanut production worldwide was 53.6 million tons in 2020 [16].

Additionally, phenolic compounds from the kernels and skins of peanuts are commonly extracted with solvents such as methanol, acetone, and ethanol with different levels of hydration [11,12,17,18]. Among them, ethanol is the least toxic and the most environmentally sustainable. Moreover, the optimal conditions to extract total phenolic compounds from peanut skins were determined as ethanol at relatively low temperatures using response surface methodology (RSM) [14]. RSM consists of the use of mathematical and statistical techniques to optimize the relationship between certain responses and associated variables [19]. To the best of our knowledge, there are no studies using RSM to optimize the extraction of antioxidant phenolic compounds from the kernels and skins of drought tolerant peanut genotypes.

Thus, the objective of our study was to establish optimal extraction conditions for the recovery of antioxidant phenolic compounds from the kernel and skin of different peanut genotypes, as well as to assess the germplasm based on phenolic composition, investigating a possible relationship with drought tolerance.

## 2. Materials and Methods

### 2.1. Contrasting Germplasm of Arachis Hypogaea

Fourteen contrasting peanut genotypes for drought tolerance belonging to the Peanut Breeding Program were kindly provided by Embrapa Algodão, as shown in Table 1 [2,3,5,6,20,21,22]. Seeds were obtained from multiplication fields carried out in Campina Grande-PB, Brazil (7°13′51″ S, 35°52′54″ W, 512 m) during the rainy season, between May and September 2015. After harvesting, seeds were dried until reaching 8–10% moisture. Skins were manually removed from the kernels, and both were kept at −20 °C until extraction.

Skins and kernels were ground in an analytical mill (IKA A11 Basic, Staufen, Germany). Kernels were defatted thrice by immersion in hexane (1:10; *m*/*v*) under stirring with further centrifugation at 8000× *g* (Eppendorf 5810R, Eppendorf AG, Hamburg, Germany) for 10 min at 25 °C; supernatants were removed, and the defatted residues were combined and rotaevaporated (Buchi Rotavapor R-215, Flawil, Switzerland).

### 2.2. Experimental Design by Response Surface Methodology (RSM)

In order to maximize the recovery of antioxidant phenolic compounds, an optimizing assay using RSM was carried out with seeds from drought tolerant (BR1) and sensitive (LViPE-06) genotypes. A Central Composite Rotatable Design (CCRD) of two factors and two levels (2^2^) and five central points was applied. Extracts were prepared by diluting the samples (25 mg of skins or 50 mg of defatted kernels) with ethanol + water (1:20, *m*/*v*) and keeping it in a thermostatted water bath for 30 min. The % ethanol (EtOH) and temperatures are provided in the Appendix A. All samples were then sonicated (UltraCleaner 1400A, Unique, Indaiatuba, Brazil) for 15 min and centrifuged at 7000× *g* for 10 min at 21 °C.

The dependent variables were total phenolic content (TPC) and antioxidant activity determined by deactivation activity of ABTS^•+^ and ROO^•^ radicals. Effects of the independent variables (% EtOH and temperature) were analyzed, and regression mathematical models were adjusted according to the following polynomial equation:Y = b_0_ + b_1_X + b_2_Z + b_11_ X ^2^ + b_22_Z^2^ + b_12_XZ
where Y is the response of the dependent variables, b values are regression coefficients, X and Z are decoded values to the independent variables % EtOH and temperature, respectively, b_1_X and b_2_Z are linear terms, b_11_X^2^ and b_22_Z^2^ are the quadratic terms, and b_12_XZ refers to two effects of factor interaction [23].

The fit of the experimental data to the generated polynomial equation was evaluated regarding the coefficient of determination of the regressions (R^2^) and F test. The F value (*p* < 0.05) for the lack of fit was obtained by analysis of variance (ANOVA) and used to verify the significance and model adequacy (Appendix A).

### 2.3. Total Phenolic Content (TPC)

TPC was determined according to a spectrophotometric method using Folin–Ciocalteau phenol reagent [24].

### 2.4. Identification and Quantification of Phenolic Compounds Using High-Performance Liquid Chromatography Coupled with Electrospray Ioniza-Tion-Quadrupole-Time of Flight-Mass Spectrometry (HPLC-ESI-QTOF-MS) and Using High-Performance Liquid Chromatography with Photodiode Array Detector (HPLC-PDA)

A Shimadzu chromatograph (Shimadzu Co., Tokyo, Japan) equipped with a LC-30AD quaternary pump, photodiode array detector, and a SIL-30AC self-injector was used to tentatively identify the compounds in the extracts. Separation was performed on a Phenomenex Luna C18 column (250 × 4.6 mm, 5 μm) at 30 °C. The mobile phase consisted of water:formic acid (99.75:0.25, *v*:*v*) (A) and acetonitrile:water:formic acid (80.00:19.75:0.25, *v*:*v*:*v*) (B). The flow rate of the mobile phase was 1.0 mL/min, and elution gradient was 10% of solvent B, increasing to 20% in 10 min, 30% in 20 min, 50% in 30 min, and then decreasing to 10% in 38 min.

The MAXIS 3G Bruker Daltonics high-resolution mass spectrometer (Bruker Daltonics, Bremen, Germany) was equipped with an electrospray ionization (ESI) source, operating in negative mode under the following operating conditions: *m*/*z* interval of 100–2000; resolution of 30,000; nebulizer gas at 29 psi; dry gas at 8 L/min; temperature of 200 °C; and HV of 4500 V.

Data analysis was performed using the MAXIS 3G software (version 4.3, Bruker Daltonics, Bremen, Germany), and compounds were tentatively identified by comparing the exact masses, MS/MS spectra, and molecular formulas with the database available in the literature.

To confirm compounds’ identities and in order to quantify them, a Shimadzu chromatograph system (Shimadzu Co., Tokyo, Japan) equipped with an SPD-M 10AVp photodiode array detector was used. Separation was performed on an Agilent C18 column with the same dimensions of the column used for the HPLC-ESI-QTOF-MS at 30 °C. The mobile phase, the flow rate, and the elution gradient also consisted of the same used for the HPLC-ESI-QTOF-MS. Spectra was recorded from 280 to 800 nm.

Data analysis was performed using the Class-VP^®^ software (version 6.1, Shimadzu Co., Tokyo, Japan), and compounds’ identities were confirmed by comparing the absorption spectra and retention time (RT) with authentic standards. Quantification was determined by external calibration using calibration curves of the authentic standards. Limits of detection and quantification (LOD and LOQ) were calculated, as was the linearity (R^2^) (Table 2 and Table 3).

### 2.5. Reactive Oxygen Species (ROS) Scavenging Activity

Assays to determine the deactivation activity of the 2-2′-azino-bis-(3-ethylbenzthiazoline-6-sulfonic acid) radical (ABTS^•+^), the peroxyl radical (ROO^•^) (measured by the oxygen radical absorbance capacity (ORAC) assay), the superoxide radical (O_2_^•−^) generated by the NADH/PMS system, and the hypochlorous acid (HOCl) were conducted according to Melo et al. [24]. The hydrogen peroxide (H_2_O_2_) scavenging activity was determined according to Chisté et al. [25], while the hydroxyl radical (^•^OH) scavenging activity was conducted according to Mariutti et al. [26].

### 2.6. Statistical Analysis

Data were analyzed statistically using univariate and multivariate methods. Kolmogorov–Smirnov’s normality test [27] and Bartlett’s homoscedasticity test [28] were carried out. The analysis of variance (ANOVA) was based on the following statistical model:Y_ij_ = m + Ti + E_ijk_
where Y_ijk_ is the phenotype verified on i treatment, k is repetition, m is the general average, T is the effect of the i_th_ treatment (I = 1, 2, …, t), and E_ij_ is the random error. Skott Knott test was used for mean comparisons.

Clustering of genotypes was estimated through principal component analysis (PCA). For this purpose, the original data were transformed into a set with equivalent dimensions but not correlated, representing, in decreasing order of estimation, the maximum variance contained in the population, according to the following expression:Y_ij_ = a_1_x_i1_ + a_2_x_i2_ + … + a_n_x_in_
where Yij is the principal component, x_in_ is the weighted average of the i_th_ accession relative to the n_th_ trait, and a_n_ is the eigenvector associated to the n_th_ trait.

Statistica software (version 13.0, StatSoft, Tulsa, OK, USA) was used for RSM analysis, while the GENES software (version 2013.5.1, Federal University of Viçosa, Viçosa, Brazil) [29] was used for the other statistical analysis. The level of confidence used was 95%.

## 3. Results

Results of the RSM for the dependent variables are shown in the Appendix A, and the three-dimensional response surface plots generated are shown in Figure 1. Analysis of variance, summarized in the Appendix A, showed that all regressions were significant (*p* < 0.05), because the ratio between the calculated F test and the tabulated value for F was higher than one. The quadratic terms were better than the linear terms (considering the R^2^), and thus, were used.

For peanut kernels, the highest temperatures (60 °C and 80 °C) combined with moderate to low % EtOH (35–50%, *v*/*v*) reached the highest TPC. The highest values were obtained at 40–80 °C with 35–60% EtOH (*v*/*v*) for the ABTS test for both genotypes, as well as for the ORAC test for the BR1 genotype. LViPE-06 presented two regions with the highest values for the ORAC test: 35–50% EtOH (*v*/*v*) at 40–55 °C and 70–80 °C. 

For peanut skins, the regions with the highest TPC were determined by the % EtOH (*v*/*v*), which was 35–60% EtOH (*v*/*v*) for BR1 and 40–60% EtOH (*v*/*v*) for LViPE-06. The highest scavenging activity against the ABTS radical was obtained using 40–60% EtOH (*v*/*v*) for BR1 and 40–78% EtOH (*v*/*v*) for LViPE-06, while for the ORAC test, the best scavenging activity was obtained using 30–60% EtOH (*v*/*v*) for BR1 and 60–75% EtOH (*v*/*v*) for LViPE-06. 

The lack of fit, comprising the residual and the pure error, was not significant (*p* > 0.05) for almost all mathematical models, except for TPC to the kernels of both genotypes and for the ORAC test to the BR1 kernel. Nevertheless, the coefficients of determination (R^2^) ranged from 0.85 to 0.97, indicating high percentages of variations explained by the polynomial equation (Appendix A).

The best extraction conditions for maximizing TPC and antioxidant activity were defined as: 60 °C and 35% EtOH (*v*/*v*) for kernels and 40 °C and 60% EtOH (*v*/*v*) for skins. Additional criteria to define those conditions were the lowest temperature, and consequently, the lower energy consumption, and the lowest % EtOH (*v*/*v*) within the optimum range of extraction for lower extraction costs, considering the results for all dependent variables. These conditions were used for further extraction of antioxidant phenolic compounds of all genotypes and yielded from 5.48% (BR1) to 14.61% (Senegal 57422) for skin extracts, and from 8.54% (Florunner) to 17.65% (BR1) for kernel extracts (Appendix A).

Among all data obtained from the 14 genotypes, only the results for TPC and ^•^OH scavenging activity for the skins and HOCl scavenging activity for the kernels did not show a normal distribution of errors and homogeneity of variances (Appendix A). Those results, however, were kept for demonstration purposes, due to the amplitude of the averages among the genotypes.

As expected, the total phenolic content was higher in the skins than in the kernels (Table 2 and Table 3). In the kernels, the TPC ranged from 18.56 mg GAE/g in BRS 151L7 to 28.72 mg GAE/g in BR1, both drought tolerant and Valencia type (Table 2). In the skins, the TPC means differed statistically (*p* ≤ 0.05) between the genotypes, and the highest values were achieved in materials with high or medium levels of drought tolerance: F.M407B (673.67 mg GAE/g), Senegal 55,437 (612.87 mg GAE/g), and Senegal 57,422 (608.21 mg GAE/g), all tan skin color (Table 3). The highest TPC values found for both kernels (BR1) and skins (FM407B) were statistically different (*p* < 0.05) for all other genotypes.

Two phenolic acids (caffeic acid and p-coumaric acid) and one flavonoid (rutin) were identified in kernel extracts (Table 2). Caffeic acid was characterized by the ion of *m*/*z* 179.0305 and by the fragment *m*/*z* 135 at the MS/MS spectra, which indicated the loss of CO_2_ ([M-H-44]^−^). p-Coumaric acid was identified by the ion of *m*/*z* 163.0399 and the fragment of *m*/*z* 119.0503, characterized by the loss of the CO_2_ group of the carboxylic acid. Rutin (quercetin-3-*O*-rutinoside) showed the ion of *m*/*z* 609.1454 and the corresponding fragment characteristic for quercetin (*m*/*z* 301.0336).

Rutin was quantified only in BR1 and Tatu genotypes. Caffeic acid concentration ranged from 0.14 mg/g (M407.424B) to 0.33 mg/g (Senegal 55437), while p-coumaric acid concentration ranged from 0.10 mg/g (Senegal 57422) to 0.55 mg/g (BR1).

Seven phenolic compounds were identified and quantified in skin extracts (Table 3): one phenolic acid (protocatechuic acid) and six flavonoids (catechin, procyanidin A2, quercetin, rutin, quercetin-3-β-D-glucoside, and kaempferol-3-glucoside). Protocatechuic acid showed the ion of *m*/*z* 153.0195 and the prominent fragment *m*/*z* 109.0297 at the MS/MS spectra, corresponding to the CO_2_ loss from the carboxylic acid. The ion of *m*/*z* 289.0711 suggested the molecular mass either of catechin or epicatechin, and then catechin was confirmed by comparison with the RT of an authentic standard. Procyanidin dimer type A, later confirmed by an authentic standard as procyanidin A2, was characterized by the ion of *m*/*z* 575.1200 and the fragment *m*/*z* 289.0722 at the MS/MS spectra [30]. Quercetin-3-β-glucoside (isoquercetin) showed the ion of *m*/*z* 463.0892 and, at the MS/MS spectra, the prominent fragments *m*/*z* 301.0323 and 300.0281 caused by, respectively, the hexose loss and quinone formation, by homolytic cleavage of the *O*-glycosidic bond [31]. Kaempferol-3-glucoside (astragalin) was tentatively identified by the precursor ion of *m*/*z* 447.0949, which was very similar to the compound previously identified by Juliano et al. [2], with the ion of *m*/*z* 447.0932, and later confirmed with an authentic standard.

Among them, three were found in the skins of all 14 genotypes (protocatechuic acid, quercetin, and rutin). Interestingly, only three genotypes (BR1, Tatu, and M407.424B), which are red skinned, contained all phenolic compounds identified in our study.

Procyanidin A2 was found in higher concentrations in skin extracts, ranging from 1.26 mg/g (FM.424B) to 7.93 mg/g (L7 Bege). (+)-Catechin was the second phenolic found in higher concentrations, ranging from 0.59 mg/g (Senegal 57422) to 3.33 mg/g (Senegal 55437). 

Quercetin-3-β-D-glucoside was found only in the genotypes with red skin, in concentrations ranging from 0.82 mg/g to 1.33 mg/g. Kaempferol-3-glucoside was found in extracts of red-skin genotypes, as well as in Senegal 57422 and LViPE-06. Higher concentrations of protocatechuic acid (0.67–0.83 mg/g) and quercetin (0.58–0.78 mg/g) were found in the red-skin genotypes BR1, Tatu, and M.407.424B, when compared with the other genotypes: 0.06–0.25 mg/g for protocatechuic acid and 0.02–0.10 mg/g for quercetin.Table 4 shows the ROS scavenging activity of the extracts. For skin extracts, the drought tolerant Senegal 55437 showed the best scavenging activities against O_2_^•−^ (IC_50_ of 12.23 µg/mL), HOCl (IC_50_ of 1.67 µg/mL), and ^•^OH (IC_50_ of 0.05 µg/mL). In contrast, the drought sensitive LViPE-06 genotype showed the worst results to scavenge those ROS: IC_50_ of 32.89 µg/mL for O_2_^•−^, 2.45 µg/mL for HOCl, and 0.10 µg/mL for ^•^OH. Nevertheless, Senegal 55437 was statistically different from other genotypes only for scavenging O_2_^•−^ (*p* < 0.05), while LViPE-06 did not differ statistically from other genotypes in any assay.

The ROO^•^ scavenging activities ranged from ca. 3100 µmol TE/g for the LViPE-06 and LGoPE-06 to ca. 5000 µmol TE/g for the FM407B, M407.424B, and Senegal 55437, which were the best results.

The best IC_50_ to scavenge H_2_O_2_ was observed for the BR1 and Tatu (ca. 25 µg/mL), and the worst was observed for the LViPE-06, LGoPE-06, and Florunner (ca. 40 µg/mL).

For kernel extracts, the scavenging activity against O_2_^•−^ was measured by inhibition percentage and not as IC_50_, because there was a decrease on inhibition percentage in concentrations >50 µg/mL (BR1 and Senegal 57422) or >100 µg/mL (other genotypes), when compared with gallic acid. BR1 showed the higher O_2_^•−^ inhibition percentage (28.65%) at 50 µg/mL, which was statistically different from the others (*p* < 0.05).

The effects of kernel extracts against H_2_O_2_ were expressed as IC_25_, because most of the extracts did not reach inhibition percentages >50% using the higher tested concentration (2000 µg/mL). The better IC_25_ was shown by LViPE-6 (228.32 µg/mL). Among the drought tolerant genotypes, the BR1 showed the best IC_25_ (304.61 µg/mL).

The kernel extract of FM407B stands out for scavenging HOCl, with an IC_50_ of 11.09 µg/mL, followed by other genotypes belonging to the same Virginia botanic type (FM407B, LGoPE-06, FM.424B, M407.424B, and Florunner) and by Senegal 57422 (the only drought tolerant), which ranged from 13.90 to 16.50 µg/mL.

The drought tolerant genotypes Senegal 55437 and BR1, respectively, showed the best results for scavenging ^•^OH (IC_50_ of 2.76 µg/mL) and ROO^•^ (738.97 µmol TE/g) at a level of confidence of 5%.

Principal component analysis was carried out using TPC, ROS scavenging activity, and phenolic composition for genotype clustering, based on their skin and kernel extracts. The joint TPC/ROS scavenging activity data, as well as only ROS scavenging activity data, did not allow coherent clustering. However, phenolic composition data from skin extracts was quite contributive, allowing a plausible clustering of genotypes, considering the genetic basis of the germplasm displayed in Table 1.

The pairwise score plot of genotypes clustered from data of skin extracts, based on protocatechuic acid, (+)-catechin, procyanidin A2, quercetin, and rutin concentrations are shown in Figure 2. The accumulated variation for the first two principal components (PC1 and PC2) corresponded to 96% of the total variance, indicating that most variability of the evaluated traits were summarized in these two components.

## 4. Discussion

In our study, the parameters for the experiment using the response surface methodology had different effects on phenolic compound extraction, depending on the seed part. While heating was important for the kernels, which is expected, because it increases compound solubility within the solvent [24], it had no effects (*p* > 0.05) for the skins. Furthermore, the optimal extraction conditions for TPC and antioxidant activity of the skins, which were 40 °C and 60% EtOH (*v*/*v*), were higher than those reported in the literature for a non-identified peanut cultivar, which were 30.9 °C and 30.8% ethanol in water [14]. Therefore, the optimal conditions to extract antioxidant phenolic compounds may vary depending on peanut cultivar/genotype, and this should be taken into account for the industrial extraction processes of phenolic compounds with health benefits from seeds.

The drought tolerant BR1 peanut genotype showed outstanding results for TPC in its kernel extract (*p* < 0.05). The TPC, as well as the levels of certain compounds, such as procyanidins, rutin, and quercetin, normally increase under water-stress conditions in medicinal and spice plants [8]. In our study, the quantitative results showed no clear increase in any individual phenolic compound in BR1 or in drought tolerant genotypes as a group. It is noteworthy, however, that all genotypes were grown during the rainy season, therefore they were not under water stress.

Concerning the skin extracts, procyanidin A was the most abundant phenolic compound. Procyanidins were also the major phenolic compounds quantified in the skin extracts of commercial Brazilian peanut genotypes [4]. The presence of the glycosidic flavonoids quercetin-3-β-D-glucoside and kaempferol-3-glucoside differentiated some of the tested genotypes, and were identified in some of the cultivars in our previous studies [2,3], and even in peanut oil from Argentina [32].

The drought tolerant genotypes Senegal 55437, BRS151 L7, and L7 Bege revealed the highest catechin concentrations, which was the second major phenolic compound in the skin extracts, while it was not detected only in drought sensitive genotypes (Table 3). It is noteworthy that Senegal 55437 is an African earliness-Spanish botanic type widely used as a parent in breeding program focusing on tolerance to drought [21,22]. In environments with water restriction and high temperature, catechin synthesis increases in drought tolerant plants [33,34], suggesting its likely role in stressful conditions. Therefore, although there was no clear association between the relatively high catechin concentration and drought tolerance (Table 3), it may be indicative of the genetic value of the progenies generated through crossings.

Furthermore, in our study, the genotypes with red skins showed a more diverse phenolic profile, including flavonoids. It is noteworthy that a previous study from our research group with shelled seeds (kernel + skin) of five genotypes (Senegal 55437, BR1, FM.424B, LViPE-06, and LGoPE-06) harvested during the same season one year later were differentiated based on their flavonoid contents [2]. In contrast, while the redness of the peanut skins was strongly correlated (r^2^ > 0.7) with the TPC, it was not correlated with the total flavonoid content in a study with peanut genotypes belonging to Runner, Valencia, Virginia, and Spanish botanic types [17].

Among the phenolic compounds, polyphenols such as flavonoids have higher antioxidant activity than monophenols, because the number and position of hydroxyl and catechol groups, as well as the presence and location of insaturations, affect their biological activity [35]. This may explain the outstanding ROS scavenging activity of the skin extracts when compared to the kernel extracts (Table 4). When in excess in plants, ROS are related to abiotic stress under field conditions, including water stress, which hampers crop productivity [8]. In the human body, excessive ROS are known to cause damages in the DNA and macromolecules through oxidation mechanisms related to chronic diseases and increased risk of vascular events [4].

The skin extracts were more effective at scavenging O_2_^•−^ than extracts of unexplored Brazilian superfruit extracts, with IC_50_ ranging from 68.33 µg/mL to 1447.94 µg/mL [36]. Similarly, the IC_50_ obtained for the skin extracts of the studied genotypes to scavenge H_2_O_2_ are outstanding, especially for the BR1 and Tatu genotypes (ca. 25 µg/mL), when compared with results for extracts of lime (143 µg/mL) and açaí (259 µg/mL) [37]. The IC_50_ of peanut skin extracts for scavenging HOCl were also lower than those showed by unexplored Brazilian fruit extracts (4.41–41.11 µg/mL) [36], which means that they are greater HOCl deactivators.

Another important ROS is ^•^OH. The best results for scavenging ^•^OH obtained by the genotypes Senegal 55437, L7 Bege, and M.407.424B are comparable to the IC_50_ of 0.05 µg/mL obtained for the medicinal Wagatea spicata leaf extracts [38]. Moreover, the ROO^•^ scavenging activity shown by the genotypes Senegal 55437, FM407B, M.407.424B, and Tatu (ca. 5000 µmol TE/g) were comparable to ascorbic acid (5400 µmol TE/g). Hence, our study confirms the potential use of peanut skins as a functional ingredient with health benefits for the industry, regardless of the cultivar/genotype.

The principal component analysis was carried out with the phenolic composition of skin extracts clustered genotypes with similar botanical and physiological traits within four robust groups. Group 1 (G1) contained the earliness genotypes: Senegal 55437, BRS 151 L7, and L7 Bege, being the first African cultivar widely used in peanut improvement as parental for drought tolerance genes [10,21]. BRS 151 L7 and L7 Bege are descendants from Senegal 55437 and IAC Tupã (a Brazilian genotype) and are very tolerant to environments with low water availability [20,39]. Group 2 (G2) contained the Africans L50 and its parent Senegal 57422, both Spanish type, tan skin, and drought tolerant.

Most of the Runner genotypes were clustered in Group 3 (G3): LViPE-06, LGoPE-06, Porto Alegre, Florunner, and FM 424B. These genotypes are sensitive to drought but are outstanding for pod production and oilseed extraction [7,36]. Group 4 (G4) was formed by Tatu and M404.424B, both red skinned and originating from Argentina. Tatu was spread in Brazil in the 1940s’ and is a parent to several Brazilian peanut cultivars.

FM424B and BR1 remained isolated in the pairwise score plot. FM424B and FM407B are isolines descendants from Florunner and differ because FM424B was mid-tolerant in field trials [40]. BR 1 is an early Valencia type widely grown on Brazilian semiarid land, being very tolerant to drought [6,7,41].

Concerning the peanut kernel, which corresponds to more than 90% of the seed, the best results shown by the extract of BR1 for O_2_^•−^% inhibition and ROO^•^ scavenging activity, and of Senegal 55437 for scavenging ^•^OH, both drought tolerant genotypes, confirm the findings of Juliano et al. [3] for the whole seeds of those drought tolerant genotypes. Conversely, the greater results for scavenging HOCl were shown mainly by drought sensitive Virginia botanic-type genotypes (FM407B, LGoPE-06, M.407.424B, and FM.424B). Those results find parallels with a study where peanut extracts from Virginia botanic type were better ABTS^•+^ scavengers than those from the Spanish botanic type [13].

Our results add biological value, especially to the BR1 and Senegal 55437 cultivar, which are earlies and are tolerant to drought [6,7]. In environments that comprise the Brazilian semiarid region, BR 1 is particularly recommended due to the high nutritional quality of the grains, which are rich in proteins, flavonoids, α-tocopherol, and polyunsaturated fatty acids [3,40]. Furthermore, the genetic isolation of these cultivars can be helpful for further breeding works focusing on peanut health benefits.

## 5. Conclusions

In conclusion, the optimization of extraction conditions of antioxidant phenolic compounds from the skin and kernel of peanut genotypes with contrasting tolerance to drought allowed the identification and quantification of phenolic acids and glycosidic and non-glycosidic flavonoids in peanut. Principal component analysis for skin extracts showed a clustering trend of the studied peanut genotypes when grown under natural field conditions, based on drought tolerance and on botanical type of germplasm. However, drought tolerant genotypes showed outstanding results, especially BR1, when the kernels were evaluated. These findings add value to BR1, which is short cycle and largely adapted to semiarid environments, and demonstrate the status quo for bioactive phenolic compounds of genotypes contrasting for drought tolerance when cultivated under natural field conditions. Further studies using seeds from those genotypes grown under water-stress conditions should be encouraged.

## Figures and Tables

**Figure 1 foods-11-00449-f001:**
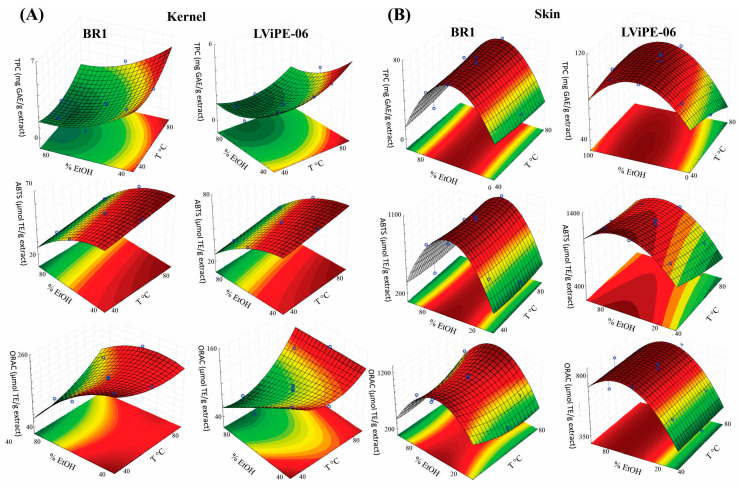
Three-dimensional response surface plots generated for kernels (**A**) and skins (**B**) of BR1 and LViPE-06 peanut genotypes assessed, showing the regions of higher antioxidant activity (red). Axis x corresponds to % EtOH, axis y corresponds to the temperature (°C), and axis z corresponds to the total phenolic content (TPC) or deactivation activity of ABTS^•+^ or ROO^•^ radicals (ABTS or ORAC, respectively).

**Figure 2 foods-11-00449-f002:**
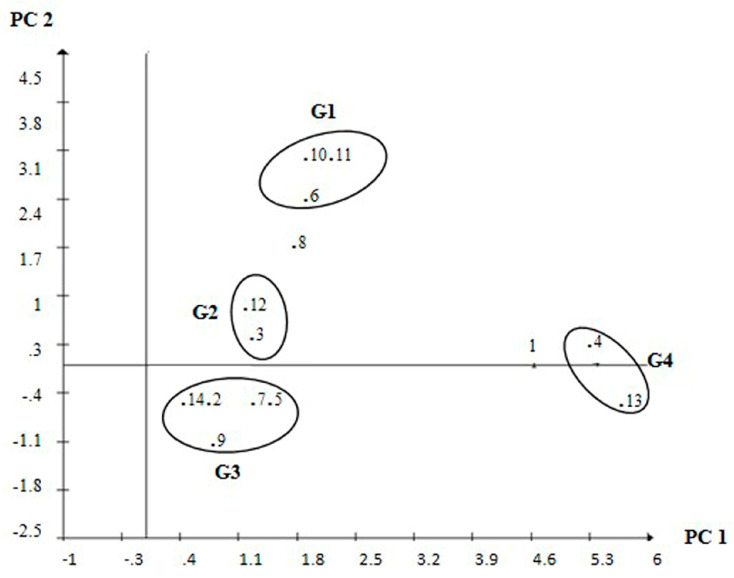
Pairwise score plot between the two first principal components (PC), based on the phenolic composition of skin extracts of different peanut genotypes. Clusters identified as G1, G2, G3 and G4. 1-BR 1, 2-LViPE-06, 3-Senegal 57422, 4-Tatu, 5-Florunner, 6-BRS 151 L7, 7-FM424B, 8-FM407B, 9-Porto Alegre, 10-L7 Bege, 11-Senegal 55437, 12-L50, 13-M407.424B, 14-LGoPE-06.

**Table 1 foods-11-00449-t001:** Traits of peanut germplasm and inheritance to drought tolerance.

Botanic Type	Genotype	Origin/Genetic Base	Skin Color	Seed Size	Cycle (Days)	Drought Tolerance
Spanish	Senegal 55437	Africa/Cultivar	Tan	Small	75–80	Tolerant
	L7 Bege	Brazil/Top line	Tan	Large	85–90	Tolerant
	Senegal 57422	Africa/Cultivar	Tan	Average	85–90	Tolerant
	L50	Africa/Top line	Tan	Average	85–90	Tolerant
Virginia	LViPE-06	Brazil/Top line	Tan	Extra large	125–130	Sensitive
	LGoPE-06	Brazil/Top line	Tan	Extra large	125–130	Sensitive
	F.M407B	Brazil/Top line	Tan	Large	110–115	Sensitive
	M407.424B	Brazil/Top line	Red	Large	110–115	Sensitive
	F.M424B	Brazil/Top line	Tan	Large	115–125	Mid tolerant
	Florunner	USA/Cultivar	Tan	Large	120–125	Sensitive
Valencia	BR1	Brazil/Cultivar	Red	Average	85–89	Tolerant
	Tatu	Argentina/Cultivar	Red	Average	95–100	Sensitive
	Porto Alegre	Brazil/Accession	Tan	Average	118–120	Sensitive
	BRS151 L7	Brazil/Cultivar	Red	Large	85–89	Tolerant

**Table 2 foods-11-00449-t002:** Total phenolic content (TPC) and quantification using HPLC-PDA of phenolic compounds in optimized kernel extracts of peanut genotypes with varying levels of drought tolerances.

Genotype	TPC (mg GAE/g Extract) *	Phenolic Profile (mg/g Extract)		
		Caffeic Acid	*p*-Coumaric Acid	Rutin
Senegal 55437 (DT)	21.03 ± 1.23 b	0.329 ± 0.009	0.192 ± 0.001	n.d.
L7 Bege (DT)	18.82 ± 0.30 d	0.203 ± 0.028	0.220 ± 0.018	n.d.
Senegal 57422 (DT)	22.29 ± 0.62 b	0.215 ± 0.008	0.101 ± 0.004	n.d.
L50 (DT)	19.87 ± 0.69 c	0.170 ± 0.016	0.180 ± 0.016	n.d.
LViPE-06 (DS)	21.77 ± 0.80 b	0.134 ± 0.004	0.190 ± 0.005	n.d.
LGoPE-06 (DS)	20.33 ± 0.40 c	0.163 ± 0.002	0.163 ± 0.004	n.d.
FM407B (DS)	20.00 ± 0.14 c	0.187 ± 0.013	0.240 ± 0.009	n.d.
M.407.424B (DS)	21.43 ± 0.58 b	0.136 ± 0.006	0.268 ± 0.017	n.d.
FM.424B (MDT)	21.55 ± 0.77 b	0.176 ± 0.015	0.125 ± 0.001	n.d.
Florunner (DS)	21.42 ± 1.41 b	0.208 ± 0.013	0.157 ± 0.005	n.d.
BR1 (DT)	28.72 ± 0.90 a	0.239 ± 0.003	0.555 ± 0.018	0.061 ± 0.002
Tatu (DS)	20.80 ± 0.88 b	0.143 ± 0.037	0.176 ± 0.027	0.054 ± 0.004
Porto Alegre (DS)	20.38 ± 0.48 c	0.261 ± 0.028	0.168 ± 0.011	n.d.
BRS151 L7 (DT)	18.56 ± 0.72 d	0.143 ± 0.005	0.187 ± 0.013	n.d.
LOD (µg/mL)		0.089	0.040	0.014
LOQ (µg/mL)		0.274	0.123	0.019
Linearity (R^2^)		0.9998	0.9999	0.9998

Results expressed as mean ± standard deviation. GAE: gallic acid equivalent. DT: drought tolerant. DS: drought sensitive. MDT: mid drought tolerant. LOQ: limit of detection. LOQ: limit of quantification. n.d.: not detected. * Different letters indicate statistically significant differences (Skott Knott test, *p* < 0.05).

**Table 3 foods-11-00449-t003:** Total phenolic content (TPC) and quantification using HPLC-PDA of phenolic compounds in optimized skin extracts of peanut genotypes with varying levels of drought tolerances.

Genotype	TPC (mg GAE/g Extract) *	Phenolic Profile (mg/g Extract)						
		Protocatechuic Acid	(+)-Catechin	Procyanidin A2	Quercetin	Rutin	Quercetin-3-β-Glucoside	Kaempferol-3-Glucoside
Senegal 55437 (DT)	612.87 ± 5.91 b	0.212 ± 0.007	3.33 ± 0.33	6.15 ± 0.84	0.076 ± 0.006	0.102 ± 0.004	n.d.	n.d.
L7 Bege (DT)	555.84 ± 18.17 c	0.176 ± 0.009	2.41 ± 0.19	7.93 ± 0.31	0.089 ± 0.006	0.055 ± 0.002	n.d.	n.d.
Senegal 57422 (DT)	608.21 ± 55.39 b	0.235 ± 0.011	0.59 ± 0.08	2.42 ± 0.76	0.024 ± 0.006	0.076 ± 0.004	n.d.	0.041 ± 0.001
L50 (DT)	555.89 ± 26.02 c	0.218 ± 0.026	0.79 ± 0.05	2.62 ± 0.13	0.040 ± 0.004	0.046 ± 0.002	n.d.	n.d.
LViPE-06 (DS)	467.57 ± 47.67 d	0.092 ± 0.012	n.d.	n.d.	0.036 ± 0.001	0.043 ± 0.011	n.d.	0.077 ± 0.007
LGoPE-06 (DS)	506.89 ± 15.41 d	0.059 ± 0.006	n.d.	n.d.	0.036 ± 0.001	0.042 ± 0.013	n.d.	n.d.
FM407B (DS)	673.67 ± 52.64 a	0.206 ± 0.005	1.44 ± 0.04	5.37 ± 0.70	0.103 ± 0.003	0.060 ± 0.007	n.d.	n.d.
M.407.424B (DS)	552.83 ± 24.99 c	0.675 ± 0.027	0.96 ± 0.03	3.89 ± 0.24	0.761 ± 0.032	0.379 ± 0.033	1.02 ± 0.05	0.099 ± 0.002
FM.424B (MDT)	538.14 ± 12.72 c	0.210 ± 0.004	n.d.	1.26 ± 0.48	0.055 ± 0.001	0.178 ± 0.006	n.d.	n.d.
Florunner (DS)	512.30 ± 3.87 d	0.252 ± 0.033	n.d.	1.30 ± 0.27	0.019 ± 0.001	0.171 ± 0.006	n.d.	n.d.
BR1 (DT)	545.62 ± 6.84 c	0.687 ± 0.078	1.61 ± 0.20	3.70 ± 0.72	0.580 ± 0.063	0.439 ± 0.025	0.82 ± 0.05	0.061 ± 0.007
Tatu (DS)	568.70 ± 3.84 c	0.826 ± 0.051	1.63 ± 0.12	4.66 ± 0.38	0.776 ± 0.048	0.440 ± 0.063	1.33 ± 0.19	0.104 ± 0.007
Porto Alegre (DS)	558.54 ± 26.72 c	0.229 ± 0.010	n.d.	n.d.	0.038 ± 0.001	0.074 ± 0.010	n.d.	n.d.
BRS151 L7 (DT)	538.33 ± 5.95 c	0.220 ± 0.003	3.13 ± 0.17	4.43 ± 0.40	0.088 ± 0.003	0.051 ± 0.005	n.d.	n.d.
LOD (µg/mL)		0.009	0.03	0.23	0.014	0.008	0.04	0.010
LOQ (µg/mL)		0.045	0.09	0.71	0.041	0.024	0.11	0.039
Linearity (R^2^)		0.9991	0.9999	0.9999	0.9999	0.9999	0.9999	0.9998

Results expressed as mean ± standard deviation. GAE: gallic acid equivalent. DT: drought tolerant. DS: drought sensitive. MDT: mid drought tolerant. LOQ: limit of detection. LOQ: limit of quantification. n.d.: not detected. * Different letters indicate statistically significant differences (Skott Knott test, *p* < 0.05).

**Table 4 foods-11-00449-t004:** Reactive oxygen species scavenging activities from optimized extracts of the skin and kernel of peanut genotypes with contrasting levels of tolerance to drought.

Seed Part/Genotype	O_2_^•−^	H_2_O_2_	HOCl	^•^OH	ROO^•^
	% Inhibition for Kernels or IC_50_ (μg/mL) for Skins	IC_25_ (μg/mL) for Kernels or IC_50_ (μg/mL) for Skins	IC_50_ (μg/mL)	IC_50_ (μg/mL)	(μmol TE/g)
Kernel					
Senegal 55437 (DT)	4.36 ± 0.79 h	443.23 ± 5.57 f	19.11 ± 0.57 c	2.76 ± 0.14 d	242.77 ± 12.87 e
L7 Bege (DT)	18.47 ± 0.25 d	505.32 ± 4.23 e	18.71 ± 0.10 c	5.02 ± 0.17 c	296.09 ± 1.80 d
Senegal 57422 (DT)	24.49 ± 1.91 b	397.13 ± 15.09 f	15.38 ± 0.19 d	5.79 ± 0.43 b	246.30 ± 28.41 e
L50 (DT)	5.96 ± 0.82 h	777.65 ± 32.98 c	24.24 ± 4.96 b	5.45 ± 0.44 b	163.97 ± 3.22 g
LViPE-06 (DS)	19.98 ± 1.33 c	228.32 ± 15.08 h	17.30 ± 0.61 c	4.82 ± 0.17 c	354.60 ± 32.17 c
LGoPE-06 (DS)	8.46 ± 1.15 g	656.18 ± 19.98 d	13.90 ± 1.24 d	5.03 ± 0.29 c	325.44 ± 0.52 d
FM407B (DS)	15.39 ± 0.95 e	870.23 ± 23.34 a	11.09 ± 0.73 e	4.36 ± 0.32 c	160.37 ± 26.54 g
M.407.424B (DS)	13.66 ± 0.80 f	760.11 ± 12.53 c	16.03 ± 0.65 d	3.93 ± 0.40 c	279.57 ± 21.62 e
FM.424B (MDT)	21.43 ± 0.21 c	631.92 ± 46.76 d	15.59 ± 0.76 d	4.58 ± 0.46 c	241.88 ± 25.16 e
Florunner (DS)	17.90 ± 0.83 d	802.06 ± 42.28 c	16.50 ± 0.48 d	6.86 ± 0.77 a	249.97 ± 9.29 e
BR1 (DT)	28.65 ± 0.18 a	304.61 ± 28.50 g	18.44 ± 0.46 c	4.69 ± 0.15 c	738.97 ± 6.89 a
Tatu (DS)	21.04 ± 3.08 c	430.76 ± 36.16 f	33.63 ± 2.10 a	4.42 ± 0.06 c	401.49 ± 22.24 b
Porto Alegre (DS)	7.57 ± 0.14 g	820.01 ± 16.95 b	20.16 ± 1.57 c	6.70 ± 0.69 a	205.20 ± 28.74 f
BRS151 L7 (DT)	12.68 ± 0.83 f	771.94 ± 13.44 c	21.22 ± 0.01 c	4.15 ± 0.67 c	234.08 ± 20.92 e
Skin					
Senegal 55437 (DT)	12.23 ± 0.49 h	38.96 ± 1.93 b	1.68 ± 0.16 c	0.050 ± 0.011 e	5151.17 ± 3.76 a
L7 Bege (DT)	17.95 ± 1.52 e	38.66 ± 1.34 b	1.67 ± 0.09 c	0.060 ± 0.001 e	4542.50 ± 313.92 c
Senegal 57422 (DT)	19.48 ± 0.94 d	31.79 ± 2.15 c	2.12 ± 0.19 b	0.067 ± 0.003 d	4072.05 ± 44.26 c
L50 (DT)	16.31 ± 0.43 f	37.76 ± 2.91 b	1.88 ± 0.02 c	0.077 ± 0.005 c	4371.66 ± 11.63 b
LViPE-06 (DS)	32.89 ± 2.24 a	40.14 ± 1.92 a	2.45 ± 0.19 a	0.104 ± 0.018 a	3108.93 ± 237.57 f
LGoPE-06 (DS)	27.99 ± 0.87 b	42.84 ± 2.08 a	2.00 ± 0.08 b	0.103 ± 0.024 a	3166.66 ± 118.59 f
FM407B (DS)	14.40 ± 0.16 g	38.23 ± 2.14 b	1.81 ± 0.11 c	0.064 ± 0.004 d	4906.55 ± 94.01 a
M.407.424B (DS)	16.26 ± 1.40 f	29.07 ± 0.93 c	2.17 ± 0.08 b	0.056 ± 0.002 e	5093.12 ± 171.13 a
FM.424B (MDT)	16.52 ± 0.86 f	36.46 ± 2.19 b	2.01 ± 0.19 b	0.071 ± 0.007 c	4465.02 ± 219.20 b
Florunner (DS)	21.10 ± 0.36 d	42.18 ± 0.17 a	1.73 ± 0.09 c	0.095 ± 0.002 b	3754.35 ± 64.26 d
BR1 (DT)	17.83 ± 1.13 e	25.89 ± 2.75 d	2.08 ± 0.04 b	0.077 ± 0.002 c	4666.06 ± 125.27 b
Tatu (DS)	14.41 ± 0.25 g	24.63 ± 1.90 d	1.70 ± 0.01 c	0.073 ± 0.001 c	5230.78 ± 65.56 a
Porto Alegre (DS)	23.64 ± 1.08 c	38.58 ± 0.11 b	2.27 ± 0.27 a	0.089 ± 0.005 b	3444.22 ± 35.54 e
BRS151 L7 (DT)	15.39 ± 0.84 f	37.44 ± 0.97 b	1.74 ± 0.04 c	0.065 ± 0.001 d	4604.19 ± 225.37 b

Results expressed as mean ± standard deviation. DT: drought tolerant. DS: drought sensitive. MDT: mid drought tolerant. TE: Trolox equivalent. Different letters within the same column indicate statistically significant differences (ANOVA and Skott Knott test, *p* < 0.05). Positive controls: gallic acid (inhibition = 85.59% and IC_50_ = 11.16 μg/mL for O2^•^ scavenging activity and IC_50_ = 0.04 μg/mL for ^•^OH scavenging activity); Ascorbic acid (IC_25_ = 92.26 and IC_50_ = 229.04 μg/mL for H_2_O_2_ scavenging activity); Quercetin (IC_50_ = 0.23 μg/mL for HOCl scavenging activity).

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
