# Peer review of "Optimizing Procedures for Antioxidant Phenolics Extraction from Skin and Kernel of Peanuts with Contrasting Levels of Drought Tolerance"

_foods, 2022, doi:10.3390/foods11030449_

Round 1

Reviewer 1 Report

There are several fixes that need to be addressed, there are major errors in the text and tables.

Enter the detector parameters in section 2.4
Line 120: Rewrite the size of the column with the first value being the length
Line 121: rewrite the percentages of the volumes with ":" and not "/" and order them from highest to lowest
Line 128, 147 and 167: Correct R2 with R2
Line 133, 239: Correct H2O2 with H2O2 
Line 132, 215, 218, 2020, 234, 237, 319, 352: Correct (O2 • -) with (O2 • -)
tables: Linearity (R2) must be reported with 4 decimals
Tables: In some tables the standard deviation is reported as ± 0.00, insert 3 decimals

Reviewer 2 Report

Your study presents a intereste to the re readers but, the paper should be reduced in size.Tables and figures sometimes show the same result one of them must be ommited. PCA analysis must be more clear.

The two phenolics detected are not new.May be are detected for the first time in Peanuts.  Botanical Names should be in italics.The origin of all these genotypes and the commercial names should be indicated etc etc 

Reviewer 3 Report

Kernel and skin of 14 peanut genotypes were studied for their total phenolic composition and antioxidant activity towards ROS generation. As authors mentioned, little is known about the effect of drought tolerance on phenolic profile and antioxidant activity in different edible parts of peanut genotypes. The paper has a clear objective and is overall well written. I have some suggestions to improve the paper:

  1. Introduction: to justify the study of peanuts kernel and skin, authors should provide some numbers on the worldwide production of peanuts as well as the % composition of kernels and skin. It is needed to justify its use in food and cosmeceutical applications. Some inspirations (not from this reviewer) that authors could base their work:

https://www.ncbi.nlm.nih.gov/pmc/articles/PMC6436756/

https://pubs.acs.org/doi/10.1021/jf8030925 This paper used methanol, water and different extracting temperatures – this should be addressed in the introduction and R&D

  1. Methods: a. the particle size of milled kernels and skin should be added
  2. authors should pay attention to the squared numbers in the text - A Central Composite Rotatable Design (CCRD) of two factors and two levels (22), R2, (O2•-, ABTS•+); section 2.4. data on analytical curves should be added in the paper.

Statistical analysis: authors are not much aware of the statistical analysis. The writing of the methodology is not correct. Authors should provide proof of homocedasticity and describe the type of ANOVA used in the study.

As for the RSM, there is no description on how the modelling was performed, which does not enable future researchers to repeat the work. As for PCA, no description was made – authors should provide details on the methodology. In addition, PCA cannot be used for classification as it is a nonsupervised methodology. If authors want to classify samples, supervised methods, such as KNN, SIMCA, and PLS-DA should be used.

  1. Results: Table 2 should contain the standard deviation for each response. If analyses were not assayed in triplicate, the paper should be rejected. D. TPC unit: TPC (µg GAE/mg)… shouldn’t it be TPC (µg GAE/g) to be in-line with ABTS and ORAC units?

Table 3 is useless – authors should create a table containing the regression coefficients, p values, confidence intervals for the coefficients. Authors should also add the adjusted R2… QF, df, QM values should be deleted as they are useless.

Reviewer 4 Report

This manuscript titled “Optimizing procedures for antioxidant phenolics extraction from skin and kernel of peanuts with contrasting levels of drought tolerance”. The comments for this manuscript are as follows:

  1. The authors should state in the section of "Materials and Methods" how many peanut skins (weight) were involved in this experiment, and what was the total extraction yield of each sample?
  2. The equation: Y = b0 + b1X + b2Z + b11X2 + b22Z2 + b12XZ . Is there a reference? Otherwise, what credibility does this equation have?
  3. Please carefully check the numbers in this manuscript. If the number exceeds 3 digits, a comma should be added. This is the real scientific number representation. The authors had omitted some places, such as line 86, 98….
  4. There should be a space between the text and the references, and the authors should correct it thoroughly and carefully.
  5. Table 4 and Table 5 are similar suas as "Senegal 55437 (DT)". Why is its TPC (mg GAE/g extract)* different? Other Genotypes have similar problems. Can the authors explain more clearly? Otherwise readers may be confused. In addition, this manualscript has too many tables, can the authors think of a way to simplify it? I don't think so many tables are needed. If the experimental results are not very important, I suggest putting them in the "Supporting Information".
  6. Can the resolution of Figure 1 be enhanced? Because it is not very clear after zooming in.
  7. The authors claims to have discovered two glycosidic flavonoids in peanut for the first time, quercetin-3-β-D-glucoside and kaempferol-3-glucoside. Is there any NMR spectrum data to prove the structure identification? Otherwise, what instrument did the authors rely on to identify its structure?

Reviewer 5 Report

The authors of this manuscript present an interesting research study on optimizing procedures for antioxidant phenolics extraction from skin and kernel of peanuts with contrasting levels of drought tolerance. Introduction and material and methods are well described. Also, results are presented in tables and figures are clear and quite explicable. Due to limited research in the specific area of these peanuts I believe that discussion section is fair and the authors discuss and explain the findings of their work. The text needs very few revisions. Although research studies on antioxidant compounds of several plant species is studied well the presented research study can add further interest to the researchers worldwide.

Abstract

COMMENT:

Abstract describes sufficient the findings of this research work.

Introduction

Introduction section is well written and, in my opinion, give the appropriate information without being extended. The purpose of the research work is clearly presented.

Line 29        Arachis hypogaea      

Lines 33, 36, 38….etc        body [2–4]    (leave  gap) apply to other within the text. Please check this suggestion according to the authors instructions and apply to the text. 

Materials and Methods

Lines 83,84     please check….what do you mean sources?...Please use gap between the two lines and the table.

Lines 105-106     please leave gap

Line 133, 225       H2O2    Please check the rest of the text

Results

Figure 1     I think that figure 1 needs to be a bit more enlarged

Line 193     Please leave gap between lines 192-193

Line 214      Please leave gap between lines 213-214

Line 215      O2•- (IC50)     or     O2•- (IC50)     please check the rest of the text like line 239, 243, 248

Line 233       Please leave gap between lines 233-234

Line 257      (+)-catechin        (do you mean ±)

Figure 2       Resolution is not good

Discussion

Lines 338, 345, 350      [10,19].    [5,32]   [6,7,32].     . Please check the rest of the text.

Conclusions

The conclusion section described sufficient the findings of the research work

References

COMMENT: Please check reference list according to the author’s instruction.

Round 2

Reviewer 3 Report

Authors have corrected the paper according to my suggestions. I recommend acceptance